# Recurrence Patterns, Treatment Outcomes, and Prognostic Factors of Thymic Carcinoma: A Multicenter Study

**DOI:** 10.3390/cancers17152513

**Published:** 2025-07-30

**Authors:** Natsuo Tomita, Shunichi Ishihara, Yoshihito Nomoto, Akinori Takada, Katsumasa Nakamura, Kenta Konishi, Kohei Wakabayashi, Yukihiko Ohshima, Maho Yamada, Masayuki Matsuo, Masaya Ito, Katsuhiro Okuda, Taiki Takaoka, Dai Okazaki, Nozomi Kita, Seiya Takano, Akio Hiwatashi

**Affiliations:** 1Department of Radiology, Nagoya City University Graduate School of Medical Sciences, 1 Kawasumi, Mizuho-cho, Mizuho-ku, Nagoya 467-8601, Japan; 2Department of Radiology, Nagoya University Graduate School of Medicine, 65 Tsurumai-cho, Shouwa-ku, Nagoya 466-8550, Japan; 3Department of Radiology, Mie University, 2-174 Edobashi, Tsu 514-8507, Japan; 4Department of Radiation Oncology, Hamamatsu University School of Medicine, Handayama 1-20-1, Chuo-ku, Hamamatsu 431-3192, Japan; 5Department of Radiology, Aichi Medical University, 1-1 Yazako-Karimata, Nagakute 480-1195, Japan; 6Department of Radiotherapy, Nagoya City West Medical Center, 1-1-1 Hirate-cho, Kita-ku, Nagoya 462-8508, Japan; 7Department of Radiology, Gifu University, 1-1 Yanagito 501-1193, Japan; 8Department of Thoracic and Pediatric Surgery, Nagoya City University Graduate School of Medical Sciences, 1 Kawasumi, Mizuho-cho, Mizuho-ku, Nagoya 467-8601, Japan

**Keywords:** thymus neoplasms, thymic carcinoma, surgery, radiotherapy, multimodal treatment, stage

## Abstract

Due to the rarity of thymic carcinoma, the lack of prospective clinical trials has led to numerous database-based studies. Consequently, detailed analyses and adverse events remain largely underinvestigated. Treatment strategies for thymic carcinoma differ slightly from those for other cancers and are based more on resectability than stage. We conducted this multicenter retrospective study of thymic carcinoma to clarify its recurrence patterns, treatment outcomes, prognostic factors, and adverse events. The study examined 101 patients: 72 underwent surgery combined with radiotherapy, and 29 received definitive radiotherapy. Stage was the sole significant factor influencing all outcomes, including overall survival; treatment modality impacted only local control. Stage IV patients had similar outcomes whether they received surgery plus radiotherapy or definitive radiotherapy. This study emphasizes the importance of stage-specific, multidisciplinary strategies to optimize outcomes, especially for stage IV patients, and it recommends prospective studies for validation.

## 1. Introduction

Thymic carcinoma is even rarer than thymoma among thymic epithelial tumors (TETs) and is generally considered a more aggressive disease. The incidence rate of the condition is approximately 0.02 per 100,000 person-years [1,2]. This condition most commonly affects adults between the ages of 40 and 70. The disease may go undetected until it begins to affect the surrounding tissues of the anterior mediastinum, making early detection extremely difficult. Due to the rarity of thymic carcinoma, the absence of prospective clinical trials has resulted in numerous database-based studies [2,3,4,5,6]. Consequently, detailed analyses and adverse events (AEs) remain largely underinvestigated.

If complete resection is possible, surgical resection is the standard of care [7]. If the surgical margins are positive, postoperative radiation therapy (RT) or chemoradiotherapy (CRT) is essential [8,9,10]. For unresectable cases, RT or CRT is the preferred treatment. Treatment strategies for thymic carcinoma differ slightly from those for other cancers and are based more on resectability than stage. However, R2 resection is performed in about a quarter of surgical cases, and that percentage rises to 40% in stage IV [2,4]. R0 resection is essential for prolonging the prognosis of patients with thymic carcinoma who undergo surgery [2].

Thymic carcinomas often require extensive irradiation in areas adjacent to the heart. Since many patients are young with a high likelihood of long-term survival, physicians pay special attention to the radiation dose delivered to the heart and lungs [11,12]. Consequently, RT techniques tend to be complex to avoid damaging these normal organs in patients with thymic carcinoma. Furthermore, due to the rarity of this cancer, the recommended RT field remains unclear. This raises concerns that RT methods and precision may influence recurrence, as observed in other cancers [13,14]. Therefore, we conducted this multicenter retrospective study of thymic carcinoma to clarify the recurrence patterns, treatment outcomes, prognostic factors, and AEs in patients who received multidisciplinary treatment, including RT.

## 2. Materials and Methods

### 2.1. Study Population

This study was conducted in collaboration with seven university hospitals in Japan’s Tokai region. The study was reviewed and approved by Nagoya City University Graduate School of Medicine’s Institutional Review Board on behalf of all institutions (approval number: 60-24-0140). The ethics committee of each institution then approved the study’s conduct. As this was a retrospective observational analysis, informed consent was obtained via an opt-out website.

The patient selection criteria for this retrospective observational study were as follows: (1) patients with a confirmed diagnosis of primary thymic carcinoma between January 2004 and December 2022, (2) patients undergoing multidisciplinary treatment including RT for the newly diagnosed thymic carcinoma, and (3) patients aged 20 years or older. The following criteria were used to determine exclusion from the study: (1) patients with a histological diagnosis of thymoma, malignant lymphoma, malignant germ cell tumor, or carcinoid, and (2) patients who have opted out of the use of their information. We identified 105 patients with primary thymic carcinoma who received multidisciplinary treatment, which included RT. Of these patients, one patient who underwent palliative RT and three patients who were never followed up after treatment were excluded from the analysis. The remaining 101 patients were included in the study’s analysis. Table 1 shows a summary of patient and treatment characteristics. The subsequent stage descriptions adhere to the ninth edition of the TNM classification system [15].

### 2.2. Treatment Characteristics

#### 2.2.1. Surgery

Surgery was performed on 72 patients (71%). Of those patients, 17 (24%) received preoperative RT, while 55 (76%) received postoperative RT. The surgical margin status was determined as follows: R0 in 48 patients (67%), R1 in 18 patients (25%), and R2 in six patients (8%). Table 2 presents the treatment modality and surgical margin status by stage.

#### 2.2.2. Radiotherapy

All patients were immobilized in the supine position and underwent computed tomography (CT). Gross tumor volume (GTV) was defined as all known gross disease based on CT and/or ^18^F-fluorodeoxyglucose positron emission tomography (FDG-PET). The clinical target volume (CTV) encompassed the GTV or tumor bed, with an additional margin of 0.5–1.0 cm. Furthermore, elective mediastinal lymph node areas (i.e., elective nodal irradiation, [ENI]) were incorporated into the CTV in 23 patients (23%). The planning target volume (PTV) was defined as the CTV plus a 0.5 cm margin. All patients were treated with 6–10 MV photon beams. Most patients (n = 86, 85%) were treated with three-dimensional conformal RT (3DCRT), while the remaining 15 patients (15%) received intensity-modulated RT (IMRT). Sixty-six patients (65%) were treated with image-guided RT (IGRT), while the remaining 35 patients (35%) underwent conventional X-ray imaging for position matching. The median fractional dose was 2.0 Gy (range, 1.8–3.0), and doses were subsequently described as equivalent dose at 2 Gy (EQD2) using the linear-quadratic (LQ) model with α/β = 10 Gy. RT was administered to the PTV with a median dose of 54 Gy (range, 36–70.0). The median doses for preoperative, postoperative, and definitive RT were 40 Gy (range, 36–50), 50 Gy (range, 40–64), and 60 Gy (range, 40–70), respectively.

#### 2.2.3. Chemotherapy

Chemotherapy was administered to 57 patients (56%). The proportion of patients receiving chemotherapy by stage was as follows: 22% for stage I–II (6 of 27), 59% for stage III (26 of 44), 82% for stage IVa (9 of 11), and 84% for stage IVb (16 of 19). In 28 cases (28%), chemotherapy was administered concurrently with RT, while in 29 cases (29%), it was not administered concurrently with RT. Among patients who received chemotherapy, the majority (n = 42, 74%) received a carboplatin and paclitaxel doublet, seven (12%) received a regimen of cisplatin, cyclophosphamide, doxorubicin, and vincristine, and eight (14%) received other treatments.

### 2.3. Statistical Analysis

The duration of the follow-up period was determined from the initiation of the initial treatment modality, which could be surgery, RT, or chemotherapy. The recurrence patterns were classified according to the International Thymic Malignancy Interest Group (ITMIG) guideline [16]. Local recurrence was defined as failure within or around the primary tumor bed. Regional recurrence was defined as recurrence within the thorax other than local recurrence, including pleural and pericardial seeding. Distant metastasis was defined as failure occurring outside the thorax, including pulmonary metastasis. Local recurrence-free survival (LRFS) was calculated from the initiation of treatment to the date of local recurrence. Progression-free survival (PFS) was defined as the time from the initiation of treatment until the date of overall recurrence or death, and overall survival (OS) was calculated from the initiation of treatment until the date of last follow-up or death. The Kaplan-Meier method was employed to estimate survivals, and the log-rank test was utilized to compare survival estimates between the groups. Subsequently, we employed univariate and multivariate analysis with a Cox proportional hazards model to evaluate prognostic factors for LRFS, PFS, and OS. AEs were evaluated using the Common Terminology Criteria for Adverse Events (CTCAE) version 5.0. The chi-square test was used to compare the occurrence of AEs between groups. A variety of statistical tests were used to compare factors among the treatment groups, including the chi-square test, the Fisher’s exact test, and the T-test. All statistical analyses were performed using EZR [17], a graphical user interface for R (version 3.6.3; R Foundation for Statistical Computing, Vienna, Austria). The significance threshold was set at *p* < 0.05.

## 3. Results

### 3.1. Patient Characteristics

Table 1 presents the patients and treatment characteristics. The median age of patients was 62 years. Most patients (n = 99, 98%) exhibited a favorable performance status (PS) of 0–1. The histopathological examination of biopsy and/or surgical specimens revealed that all patients were diagnosed with primary thymic carcinoma. The histology of most patients was squamous cell carcinoma (n = 85, 84%). Only a small percentage of patients (27%) were in stage I or II, and most patients (73%) were classified as being in stage III or IV. Of the 19 patients in stage IVb, all had N2, and none had M1b (i.e., distant metastasis).

### 3.2. Outcomes and Relapse Patterns

The median follow-up period was 68 months (range, 2–220). Of the total cohort, 32 patients (32%) died, with 25 (25%) dying from thymic cancer and 7 (7%) from other causes. A total of 61 patients (60%) experienced at least one type of relapse, 18 of whom experienced two types of relapse simultaneously or heterochronously. The total number of patients who experienced local recurrence was 17 (17%), regional recurrence was observed in 27 (27%), and distant metastases were noted in 35 (35%) patients. Of the patients in this cohort, five exhibited both local and regional recurrence, five demonstrated both local recurrence and distant metastases, and eight displayed both regional recurrence and distant metastases. The sites of regional recurrence included pleural dissemination (n = 22), followed by mediastinal lymph nodes (n = 4) and pericardial dissemination (n = 3). The three most common sites of distant metastasis were the lung (n = 13), bone (n = 10), and liver (n = 10).

Figure 1A shows the survival curves for LRFS, PFS, and OS for the entire patient population. The 5-year LRFS, PFS, and OS rates were 82% (95% confidence interval [CI], 73–88), 41% (95% CI, 31–51), and 76% (95% CI, 65–84), respectively. Figure 1B–D shows the survival curves for LRFS, PFS, and OS by stage. The 5-year LRFS rates for stages I–II, III, IVa, and IVb were 96%, 84%, 82%, and 49%, respectively (*p* = 0.005). The 5-year PFS rates for stage I–II, III, IVa, and IVb were 69%, 40%, 32%, and 9%, respectively (*p* < 0.001). The five-year OS rates for stages I–II, III, IVa, and IVb were 92%, 78%, 72%, and 51%, respectively (*p* = 0.007).

### 3.3. Prognostic Factors for LRFS, PFS, and OS

Table 3 presents the results of univariate analyses for LRFS, PFS, and OS. The TNM stage was associated with all LRFS, PFS, and OS (*p* = 0.001, *p* < 0.001, and *p* < 0.001, respectively). The treatment modality was evaluated in two categories: definitive RT and surgery with preoperative or postoperative RT. The treatment modality exhibited a statistically significant correlation with overall LRFS, PFS, and OS (*p* < 0.001, *p* = 0.017, and *p* = 0.008, respectively). The evaluation of chemotherapy was conducted in three distinct categories: absence of utilization, utilization other than concurrent with RT, and concurrent utilization with RT. There is a statistically significant association between chemotherapy and PFS (*p* = 0.005).

The results of the multivariate analyses for LRFS, PFS, and OS are shown in Table 4. The TNM stage was associated with all LRFS, PFS, and OS (*p* = 0.040, *p* < 0.001, and *p* = 0.048, respectively). The treatment modality did not demonstrate a correlation with either PFS or OS and was associated with only LRFS (*p* = 0.015). IGRT and ENI were also associated with LRFS (*p* = 0.002 and 0.013, respectively). The association between ENI and PFS was also statistically significant (*p* = 0.023).

### 3.4. Outcomes by Treatment Modality According to Stage

Table 5 shows the characteristics of the patients by treatment modality. The surgery with RT group was younger than the definitive RT group (*p* = 0.044) and the tumor diameter was smaller (*p* < 0.001). PS performed well in both groups, but it was superior in the surgery group. While patient backgrounds differed between the two groups, we compared treatment outcomes between surgery with RT and definitive RT groups.

Figure 2A–C shows the comparison of LRFS between surgery with RT and definitive RT groups. Figure 2A–C show those of all patients, stages III and IV. The 5-year LRS rates of surgery with RT vs. definitive RT groups were 91% vs. 54% in all patients (*p* < 0.001), 94% vs. 50% in stage III (*p* < 0.001), and 72% vs. 57% in stage IV (*p* = 0.59), respectively.

Figure 2D–F shows the comparison of PFS between surgery with RT and definitive RT groups. Figure 2D–F shows those of all patients, stages III and IV. The 5-year PFS rates of surgery with RT vs. definitive RT groups were 46% vs. 30% in all patients (*p* = 0.015), 40% vs. 40% in stage III (*p* = 0.73), and 11% vs. 25% in stage IV (*p* = 0.99), respectively.

Figure 2G–I shows the comparison of OS between the surgery with RT and definitive RT groups. Figure 2G–I shows those of all patients, stages III and IV. The 5-year OS rates of surgery with RT vs. definitive RT groups were 85% vs. 52% in all patients (*p* = 0.006), 85% vs. 56% in stage III (*p* = 0.016), and 72% vs. 51% in stage IV (*p* = 0.98), respectively.

### 3.5. Adverse Events

Among the 72 patients who underwent surgery, 19 (26%) experienced grade 2 or higher AEs related to surgery. Of those patients, eight (11%), seven (10%), and four (6%) experienced grade 2, 3, and 4 AEs, respectively. Grade 2 AEs related to surgery included recurrent nerve palsy (n = 2), wound infection (n = 2), arrhythmia (n = 2), pleural effusion (n = 1), and keloid (n = 1). Grade 3 AEs included pleural effusion (n = 2), recurrent nerve palsy (n = 1), pericardial effusion (n = 1), atelectasis (n = 1), tracheobronchial fistula (n = 1), and pharyngeal stenosis (n = 1). Grade 4 AEs are listed below: Respiratory failure (n = 3) and cardiac tamponade (n = 1). There were no grade 5 AEs related to surgery. Grade 2 or higher AEs related to RT were observed in 14 (14%) patients. Nine of these patients experienced grade 2 AEs, and five experienced grade 3 AEs. All grade 2 RT-related AEs were radiation pneumonitis. Grade 3 RT-related AEs included radiation pneumonitis (n = 3), pleural effusion (n = 1), and pulmonary embolism (n = 1). No increase in infection rates, incidence of autoimmune diseases, and secondary malignancies was observed after treatment. There was no grade 4 or higher RT-related AEs.

The number of grade 2 or higher AEs related to surgery or RT was 21 (29%) in the surgery with RT group (n = 72) and 6 (21%) in the definitive RT group (n = 29) (*p* = 0.46). The number of grade 3 or higher AEs related to surgery or RT was 11 (15%) in the surgery with RT group and 2 (7%) in the definitive RT group (*p* = 0.34).

## 4. Discussion

This multicenter retrospective study provides significant insight into the outcomes and optimal RT approaches in multidisciplinary treatment for thymic carcinoma, a rare and aggressive malignancy with limited prospective data. The 5-year OS rate of 76% observed in our cohort highlights that multidisciplinary treatment can achieve favorable long-term control, especially in earlier stages. The 5-year OS was reported to be around 60% [2,4,18], and the results of this study were favorable. The reason is unclear, but there may have been more cases with good PS. PS is unknown in the database study [2,4,18], making comparisons difficult. PFS was lower at 41%, reflective of the high rates of regional recurrence (27%) and distant metastasis (35%), which underscore the aggressive behavior of thymic carcinoma and the challenge in achieving durable disease control. These findings are consistent with previous reports emphasizing the disease’s proclivity for early invasion and metastasis [19,20,21].

Our findings from univariate and multivariate analyses demonstrate that TNM nineth stage significantly influences LRFS, PFS, and OS, reinforcing its prognostic value. An official TNM-based system for TETs was included for the first time in the eighth edition of the TNM manual of the Union for International Cancer Control (UICC) [22,23]. Prior to the release of the TNM eighth version, the most widely used have been the Masaoka classification based on data of 91 patients, and the Masaoka-Koga classification based on data of 76 patients [24,25]. In the TNM eighth version, a retrospective database was created by ITMIG, representing the collaborative effort of 105 institutions worldwide and including 10,808 patients. It was further updated to nineth edition and modified to better correlate with prognosis [15].

Treatment modality was associated with only LRFS in multivariate analyses. Figure 2 shows that LRFS, PFS and OS are better in the surgery with RT group for all patients but are exactly the same for stage IV. This study was not designed to compare each treatment, so caution should be exercised in interpretation, as shown in Table 5. Nevertheless, these results suggest that definitive RT may be superior in stage IV, and this might be warranted to be validated in a prospective study. Several studies have suggested the benefit of debulking surgery for locally advanced thymic carcinoma [2,18]. However, these studies were not limited to stage IV disease, and it is unclear whether patients intended to undergo debulking surgery before surgery or whether surgery resulted in incomplete resection. In our current study, surgery on stage IV patients resulted in R1 and R2 resection in about half (5 of 11 patients). Due to these incomplete resections and early invasion and metastasis of thymic carcinoma, PFS and OS were comparable between the surgery with RT and definitive RT groups. On the other hand, LRFS and OS were better in the surgery group for stage III patients, although PFS was comparable. This suggests that surgery with RT may be superior to definitive RT in stage III patients. However, caution should be exercised in interpreting this case as well, since the patient backgrounds of the two groups are different. Currently, it is considered important to perform complete surgical resection of thymic carcinoma as much as possible [2,26].

Chemotherapy did not improve all outcomes in our current study. This is not consistent with findings that platinum-based regimens such as carboplatin and paclitaxel enhance efficacy in this setting, particularly when administered concurrently with RT [19,27]. This may be because many stage IV cases were treated with chemotherapy, and many cases were not concurrently treated with RT. The finding of high levels of programmed cell death ligand 1 (PD-L1) expression were confirmed in thymoma and thymic carcinoma [28]. The recent multicenter single-arm phase 2 trial showed that atezolizumab plus carboplatin and paclitaxel might become a viable treatment option for previously untreated advanced or recurrent thymic carcinoma [29]. In the future, the introduction of immune checkpoint inhibitors may improve treatment outcomes in the multidisciplinary treatment of thymic carcinoma.

Importantly, RT technique played a significant role in outcomes. The use of IGRT was independently associated with improved LRFS and PFS, suggesting that modern RT techniques allow more precise targeting while potentially mitigating local and regional failures. These results align with recent guidelines favoring advanced modalities like IMRT and IGRT to enhance dose conformity and minimize exposure to critical organs such as the heart and lungs [10,21,30,31]. The controversy over the optimal radiation field, particularly ENI, remains. ENI’s extending radiation fields may have improved PFS by reducing local recurrence or pleural dissemination rather than lymph node metastasis. The prevalence of pleural dissemination among regional recurrences suggests that extending radiation fields or using novel techniques to target these areas could be beneficial.

Resent research investigates the role of WNT4 signaling in TETs, finding that WNT4 and its receptor FZD6 are overexpressed in aggressive B3 thymoma and thymic carcinoma, unlike their age-related decline in normal thymus [32]. Unfortunately, we did not check for WNT4 overexpression in this study. As all patients included in this study had pathologically confirmed thymic cancer and no concomitant thymoma, based on the patient’s background, it is difficult to make a clinical assumption about the transition from thymoma to cancer.

The toxicity profile was acceptable though notable, with 18% experiencing ≥ grade 2 AEs due to surgery and 15% due to RT. This emphasizes the need for careful management of treatment-related morbidity, especially due to the anatomical challenges of irradiating mediastinal tumors close to vital structures [10,30]. There was no significant difference in AEs between the surgery with RT and definitive RT groups, but grade 3 or higher AEs were twice as common in the surgery with RT group (15% vs. 7%). If the outcomes are truly equivalent for both treatment modalities, RT or CRT is recommended due to medical economics and patient burden. As the thymus is an immune tissue/organ, any alteration to its function (by the cancer or its treatment) may affect overall immunity as well. However, no increase in infectious diseases or secondary cancers suggesting immune system abnormalities was observed.

Our study’s limitations include its retrospective design and potential selection bias, yet it represents one of the largest cohorts of thymic carcinoma patients receiving multidisciplinary treatment, including RT. Future research should focus on optimizing radiation parameters, investigating the role of novel systemic therapies, and developing risk-adapted treatment algorithms to improve outcomes for this challenging disease. Despite these limitations, the present results suggest the importance of stage-specific treatment strategies and need to be verified in future prospective studies.

## 5. Conclusions

Our study reinforces the pivotal role of multidisciplinary, stage-tailored treatment combining surgery, chemotherapy, and advanced RT techniques to maximize survival and local control in thymic carcinoma. In particular, among stage IV patients, the prognosis was equivalent between the group that underwent surgery combined with RT and the group that underwent definitive RT, suggesting that treatment selection based on disease stage rather than resectability is recommended for this malignant tumor. The data advocate for the integration of IGRT and extending radiation fields to reduce recurrence risks. Future prospective trials are warranted to refine these strategies, optimize systemic therapies, and improve quality of life for these patients.

## Figures and Tables

**Figure 1 cancers-17-02513-f001:**
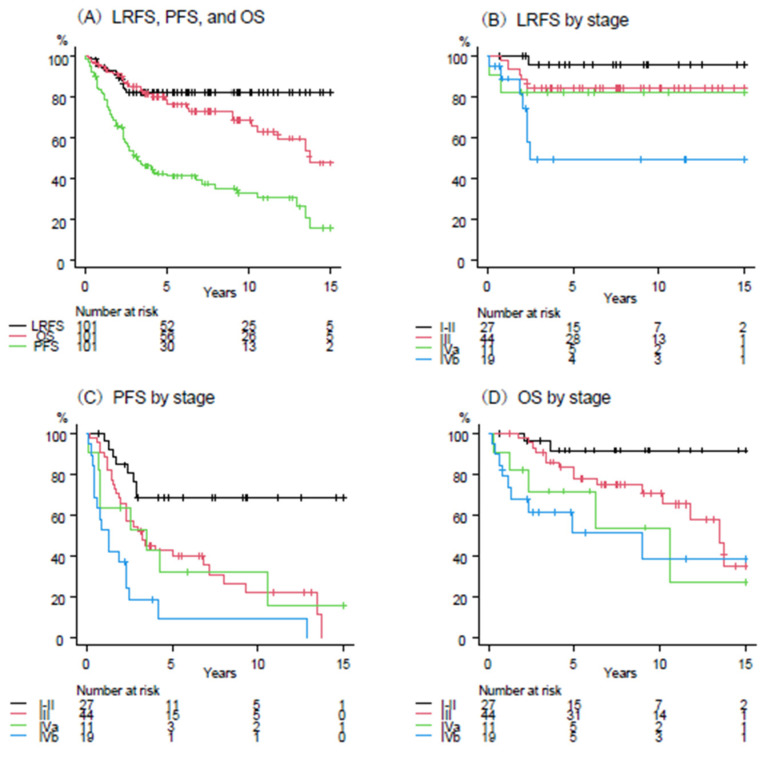
Kaplan-Meier curves of local recurrence-free survival (LRFS), progression-free survival (PFS), and overall survival (OS) for all patients (**A**), LRFS by stage (**B**), PFS by stage (**C**), and OS by stage (**D**).

**Figure 2 cancers-17-02513-f002:**
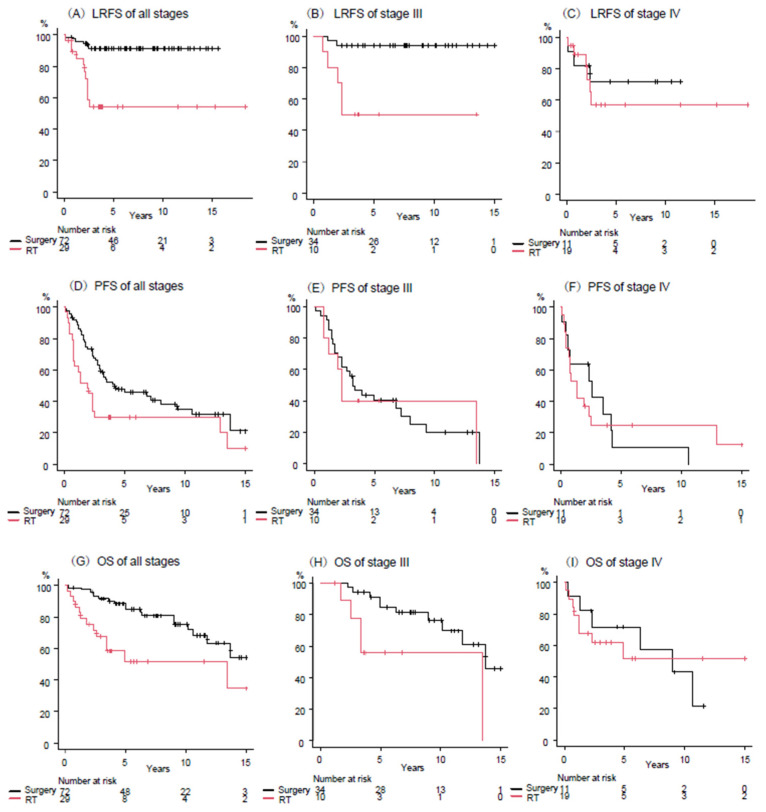
Kaplan-Meier curves of local recurrence-free survival (LRFS), progression-free survival (PFS), and overall survival (OS) for all patients (**A**,**D**,**G**), stage III (**B**,**E**,**H**), and stage IV (**C**,**F**,**I**).

**Table 1 cancers-17-02513-t001:** Patient and treatment characteristics.

Characteristic	n = 101
Age (years)	62 (24–82)
Male/female	50/51
ECOG PS 0/1/2	62/37/2
Histology by the WHO classification	
Squamous cell carcinoma	85
Thymic carcinoma, NOS	10
Neuroendocrine carcinoma	4
Undifferentiated carcinoma, NOS	1
Adenocarcinoma	1
Stage (TNM 9th)	
I/II/IIIa/IIIb/IVa/IVb	14/13/31/13/11/19
T1/2/3/4	15/15/46/25
N0/1/2	72/10/19
M0/1a/1b	98/3/0
Maximum tumor diameter (cm)	6.0 (1.8–12.0)
Treatment methods	
Surgery after preoperative RT	17
Surgery followed by postoperative RT	55
Definitive RT	29
Chemotherapy	
Use/non-use	57/44
Concurrent/non-concurrent	28/29
RT technique	
3DCRT/IMRT	86/15
Non-IGRT/IGRT	35/66
ENI/non-use	23/78
RT dose (Gy) *	54 (36–70)
Preoperative RT	40 (36–50)
Postoperative RT	50 (40–64)
Definitive RT	60 (40–70)

Data are shown as n or medians (range). * RT data were described as equivalent 2-Gy fractions (EQD2) using the linear-quadratic (LQ) model with α/β = 10 Gy. ECOG, Eastern Cooperative Oncology Group; PS, performance status; WHO, World Health Organization; NOS, not otherwise specified; RT, radiotherapy; 3DCRT, three-dimensional conformal radiotherapy; IMRT, intensity-modulated radiotherapy; IGRT, image-guided radiotherapy; ENI, elective nodal irradiation.

**Table 2 cancers-17-02513-t002:** Treatment modality and surgical margin status by stage.

		Treatment Modality					Surgical Margin Status		
		Pre-RT and Surgery	Surgery and Post-RT	Definitive RT			R0	R1	R2
I–II	(n = 27)	1 (4%)	26 (96%)	0 (0%)	I–II	(n = 27)	20 (74%)	7 (26%)	0 (0%)
III	(n = 44)	13 (30%)	21 (48%)	10 (23%)	III	(n = 34)	22 (65%)	8 (24%)	4 (12%)
IVa	(n = 11)	1 (9%)	5 (45%)	5 (45%)	IVa	(n = 6)	3 (50%)	2 (33%)	1 (17%)
IVb	(n = 19)	2 (11%)	3 (16%)	14 (74%)	IVb	(n = 5)	3 (60%)	1 (20%)	1 (20%)

RT, radiotherapy.

**Table 3 cancers-17-02513-t003:** Univariate analyses of clinical and treatment factors predictive of LRFS, PFS, and OS.

	LRFS		PFS		OS	
Predictor	HR (95% CI)	*p*-Value	HR (95% CI)	*p*-Value	HR (95% CI)	*p*-Value
Age (>65 years vs. ≤65)	1.1 (0.41–3.0)	0.84	0.76 (0.41–1.4)	0.38	1.0 (0.49–2.2)	0.93
TNM stage (IVb, IVa, III, I–II)	2.1 (1.3–3.3)	0.001	1.7 (1.4–2.2)	<0.001	1.8 (1.3–2.4)	<0.001
Treatment modality *	6.3 (2.3–17)	<0.001	1.9 (1.1–3.2)	0.017	2.6 (1.3–5.4)	0.008
Chemotherapy **	0.72 (0.41–1.3)	0.26	0.67 (0.51–0.89)	0.005	0.73 (0.49–1.1)	0.13
RT dose (≤54 Gy vs. >54 Gy)	0.76 (0.29–2.0)	0.58	0.74 (0.46–1.2)	0.24	1.2 (0.58–2.4)	0.66
RT technique (3DCRT vs. IMRT)	0.49 (0.16–1.5)	0.21	1.2 (0.53–2.6)	0.71	0.60 (0.21–1.8)	0.36
IGRT (no use vs. use)	1.9 (0.74–5.0)	0.18	1.2 (0.70–2.0)	0.53	1.2 (0.57–2.6)	0.62
ENI (no use vs. use)	2.0 (0.45–8.6)	0.37	1.0 (0.56–1.9)	0.94	0.51 (0.24–1.1)	0.070

* Treatment modality was evaluated in two categories: definitive RT vs. surgery with preoperative or postoperative RT. ** Chemotherapy was evaluated in three categories; no use vs. use other than concurrent with RT vs. concurrent use with RT. LRFS, local recurrence-free survival; PFS, progression-free survival; OS, overall survival; HR, hazard ratio; CI, confidence interval; RT, radiotherapy; 3DCRT, three-dimensional conformal radiotherapy; IMRT, intensity-modulated radiotherapy; IGRT, image-guided radiotherapy; ENI, elective nodal irradiation.

**Table 4 cancers-17-02513-t004:** Multivariate analyses of clinical and treatment factors predictive of LRFS, PFS, and OS.

	LRFS		PFS		OS	
Predictor	HR (95% CI)	*p*-Value	HR (95% CI)	*p*-Value	HR (95% CI)	*p*-Value
Age (>65 years vs. ≤65)	0.51 (0.16–1.7)	0.28	0.75 (0.41–1.4)	0.37	0.68 (0.28–1.7)	0.41
TNM stage (IVb, IVa, III, I–II)	2.1 (1.0–4.1)	0.040	1.9 (1.4–2.7)	<0.001	1.6 (1.0–2.5)	0.048
Treatment modality *	5.8 (1.4–24)	0.015	1.1 (0.56–2.2)	0.77	1.9 (0.68–5.3)	0.22
Chemotherapy **	0.96 (0.46–2.0)	0.92	0.86 (0.62–1.2)	0.39	0.94 (0.58–1.5)	0.79
RT dose (≤54 Gy vs. >54 Gy)	1.7 (0.52–5.7)	0.37	0.88 (0.50–1.5)	0.65	2.2 (0.91–5.2)	0.082
RT technique (3DCRT vs. IMRT)	0.27 (0.06–1.2)	0.083	1.3 (0.52–3.1)	0.59	0.47 (0.14–1.6)	0.24
IGRT (no use vs. use)	9.9 (2.3–43)	0.002	1.6 (0.86–2.9)	0.14	1.7 (0.71–4.2)	0.23
ENI (no use vs. use)	8.3 (1.6–43)	0.013	2.4 (1.1–4.9)	0.023	0.84 (0.33–2.1)	0.71

* Treatment modality was evaluated in two categories: definitive RT vs. surgery with preoperative or postoperative RT. ** Chemotherapy was evaluated in three categories; no use vs. use other than concurrent with RT vs. concurrent use with RT.

**Table 5 cancers-17-02513-t005:** Patient and treatment characteristics by treatment modality.

Characteristic	Surgery with RT	Definitive RT	*p*-Value
Age (years)	60 (24–82)	64 (27–82)	0.044
Male/female	35/37	15/14	0.78
ECOG PS 0/1/2	50/21/1	12/16/1	0.008
Histology			
Squamous cell carcinoma	61	24	1
Others	11	5	
Stage I–II/III/IVa/IVb	27/34/6/5	0/10/5/14	<0.001
Maximum tumor diameter (cm)	5.1 (1.8–12.0)	8.0 (3.5–12.0)	<0.001

Data are shown as n or medians (range). ECOG, Eastern Cooperative Oncology Group; PS, performance status.

## Data Availability

The datasets generated and/or analyzed during the present study are not publicly available due to ethical reasons but are available from the corresponding author upon reasonable request.

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
