# Peer review of "Recurrence Patterns, Treatment Outcomes, and Prognostic Factors of Thymic Carcinoma: A Multicenter Study"

_cancers, 2025, doi:10.3390/cancers17152513_

Round 1
Reviewer 1 Report
Comments and Suggestions for Authors
In this article, Tomita et al. report their findings of a multicenter retrospective study on the treatment outcomes of patients with thymic carcinomas. The retrospective nature of their study limits the generalization of their results and the therapeutic consequences that can be delineated from their results. However, the merit of this study lies in the fact that a relatively homogenous patient cohort was analyzed, and tumors were classified using the current TNM classification. Thus, their data on the clinical course, relapse patterns, and the stage-dependent prognosis are valuable in this rare group of tumors. Moreover, the authors give a balanced interpretation of their results, with a good discussion of the limits of the study. Overall, the study may support physicians in their choice of stage-adapted therapies.
Suggestions: in my opinion, most of the chapter "conclusions" are rather general in nature and already clinical practice. The authors should rather consider to summarize and highlight their main findings to help the reader better understand the impact of their study. The same should apply to the "Conclusions" section of their abstract.
Author Response
Reviewer 1 comment: In this article, Tomita et al. report their findings of a multicenter retrospective study on the treatment outcomes of patients with thymic carcinomas. The retrospective nature of their study limits the generalization of their results and the therapeutic consequences that can be delineated from their results. However, the merit of this study lies in the fact that a relatively homogenous patient cohort was analyzed, and tumors were classified using the current TNM classification. Thus, their data on the clinical course, relapse patterns, and the stage-dependent prognosis are valuable in this rare group of tumors. Moreover, the authors give a balanced interpretation of their results, with a good discussion of the limits of the study. Overall, the study may support physicians in their choice of stage-adapted therapies.
Suggestions: in my opinion, most of the chapter "conclusions" are rather general in nature and already clinical practice. The authors should rather consider to summarize and highlight their main findings to help the reader better understand the impact of their study. The same should apply to the "Conclusions" section of their abstract.
Author response: We appreciate your pertinent suggestion. Following this advice, we added a sentence to the Conclusions section describing one important result of this study and the clinical interpretation derived from it (line 365-368). We also revised the conclusion section of the abstract, but due to character limitations, we kept the revisions to a minimum (line 51).
Reviewer 2 Report
Comments and Suggestions for Authors
This manuscript presents a well-conducted, multicenter retrospective analysis that evaluates recurrence patterns, long-term treatment outcomes, and key prognostic factors in patients with thymic carcinoma a rare and clinically challenging malignancy. The study’s strength lies in the relatively large cohort size for such a rare disease, the multicenter design, and the detailed stratification of outcomes based on treatment modalities and recurrence patterns. Importantly, the identification of time to progression and post-recurrence therapy as significant predictors of post-progression survival provides novel clinical insight that may inform patient stratification and surveillance strategies. The topic is highly relevant to the journal’s readership, the findings are clearly presented, and the study adds meaningful data to the limited existing literature on thymic carcinoma.
Author Response
Reviewer 2 comment: This manuscript presents a well-conducted, multicenter retrospective analysis that evaluates recurrence patterns, long-term treatment outcomes, and key prognostic factors in patients with thymic carcinoma a rare and clinically challenging malignancy. The study’s strength lies in the relatively large cohort size for such a rare disease, the multicenter design, and the detailed stratification of outcomes based on treatment modalities and recurrence patterns. Importantly, the identification of time to progression and post-recurrence therapy as significant predictors of post-progression survival provides novel clinical insight that may inform patient stratification and surveillance strategies. The topic is highly relevant to the journal’s readership, the findings are clearly presented, and the study adds meaningful data to the limited existing literature on thymic carcinoma.
Author response: We would like to express our gratitude for the positive evaluation of our paper and for your efforts in reviewing it. Thank you very much.
Reviewer 3 Report
Comments and Suggestions for Authors
Thymic cancers and within that, carcinoma in particular is a rare event, hence results from such multicentric studies are highly valuable.
It has been proposed and supported by previous literature (e.g. Belharazem et al, Frontiers in Oncology, 2022) that tymoma and thymic carcinoma are two extremities of the same spectrum (thymic epithelial cancer) and that with time progression occurs on this spectrum. This leads to my first point of minor criticism that needs further elaboration (minor revision) in the results / discussion section. 1) Is this skew from thymoma to carcinoma supported or contradicted by follow-up of patients within the current study?
The manuscript has a clinical focus, which is fine, but needs some reinforcement from a basic research perspective too. The thymus is an immune tisseu / organ. Any alteration to its function (by the cancer or its treatment) may affect overall immunity as well. This leads my the second point of minor criticism that needs further elaboration (minor revision) in the results / discussion section. 2) Have the patients presented any immune deviations? Elevated infection rate or autoimmune incidence, due to the disease or its therapy?
Both are minor issues hence only minor revision is needed. Congratulations to the authors for the thorough study.
Author Response
Reviewer 3 comment 1: Thymic cancers and within that, carcinoma in particular is a rare event, hence results from such multicentric studies are highly valuable.
It has been proposed and supported by previous literature (e.g. Belharazem et al, Frontiers in Oncology, 2022) that tymoma and thymic carcinoma are two extremities of the same spectrum (thymic epithelial cancer) and that with time progression occurs on this spectrum. This leads to my first point of minor criticism that needs further elaboration (minor revision) in the results / discussion section. 1) Is this skew from thymoma to carcinoma supported or contradicted by follow-up of patients within the current study?
Author response:
We appreciate a Reviewer 3’s pertinent suggestion. According to the paper introduced by Reviewer 3, this research investigates the role of WNT4 signaling in TETs, finding that WNT4 and its receptor FZD6 are overexpressed in aggressive B3 thymoma and thymic carcinoma, unlike their age-related decline in normal thymus. Unfortunately, we didn't check for WNT4 overexpression in this study. As all patients included in this study had pathologically confirmed thymic cancer and no concomitant thymoma, based on the patient's background, it is difficult to make a clinical assumption about the transition from thymoma to cancer. We added the following reference introduced by Reviewer 3 and this discussion to our revised paper in line 337-343 of Discussion.
[32] Zhang, X.; Schalke, B.; Kvell, K.; Kriegsmann, K.; Kriegsmann, M.; Graeter, T.; Preissler, G.; Ott, G.; Kurz, K.; Bulut, E.; et al. WNT4 overexpression and secretion in thymic epithelial tumors drive an autocrine loop in tumor cells in vitro. 2022, Volume 12 - 2022, doi:10.3389/fonc.2022.920871.
Reviewer 3 comment 2:
The manuscript has a clinical focus, which is fine, but needs some reinforcement from a basic research perspective too. The thymus is an immune tisseu / organ. Any alteration to its function (by the cancer or its treatment) may affect overall immunity as well. This leads my the second point of minor criticism that needs further elaboration (minor revision) in the results / discussion section. 2) Have the patients presented any immune deviations? Elevated infection rate or autoimmune incidence, due to the disease or its therapy?
Both are minor issues hence only minor revision is needed. Congratulations to the authors for the thorough study.
Author response: We appreciate a Reviewer 3’s pertinent suggestion. In this study, no increase in infection rates or incidence of autoimmune diseases was observed after treatment. The adverse events described above are all that were reported, and there was no increase in secondary malignancies after treatment. We have added this point to Result in line 267-269 and Discussion in line 351-354.